# “I Need Someone to Help Me Build Up My Strength”: A Meta-Synthesis of Lived Experience Perspectives on the Role and Value of a Dietitian in Eating Disorder Treatment

**DOI:** 10.3390/bs13110944

**Published:** 2023-11-17

**Authors:** Yive Yang, Janet Conti, Caitlin M. McMaster, Milan K. Piya, Phillipa Hay

**Affiliations:** 1Translational Health Research Institute, School of Medicine, Western Sydney University, Campbelltown, NSW 2560, Australia; yive.yang@gmail.com (Y.Y.); j.conti@westernsydney.edu.au (J.C.); m.piya@westernsydney.edu.au (M.K.P.); 2School of Psychology, Western Sydney University, Penrith, NSW 2750, Australia; 3Sydney School of Health Sciences, Faculty of Medicine and Health, University of Sydney, Camperdown, NSW 2050, Australia; caitlin.mcmaster@sydney.edu.au; 4Eating Disorder and Nutrition Research Group (ENRG), Translational Health Research Institute, Faculty of Medicine, Western Sydney University, Campbelltown, NSW 2560, Australia; 5South Western Sydney Local Health District, Liverpool, NSW 2170, Australia; 6Ingham Institute for Applied Medical Research, Liverpool, NSW 2170, Australia; 7Camden and Campbelltown Hospitals, South Western Sydney Local Health District, Liverpool, NSW 2560, Australia

**Keywords:** feeding and eating disorders, dietitian, lived experience, dietetic counselling, nutrition therapy, qualitative review, metasynthesis

## Abstract

Dietitians are included in eating disorder (ED) treatment teams for their expertise in nutrition. However, little is known about an individual’s experience of dietetic intervention as part of their ED treatment and what they value as part of dietetic care. Therefore, the aim of this review was to synthesise the available qualitative literature to understand the role and value of a dietitian in ED treatment from the perspective of individuals with lived experience. Six databases and Google Scholar were searched and a thematic synthesis and meta-synthesis of fifteen studies were conducted. Four themes were constructed from the data: (1) “guidance and structure”—Provision of nutrition knowledge and skills; (2) “having all my bases covered”—Dietitians as part of a multidisciplinary team; (3) Challenges in nutritional treatment; and (4) “it was my treatment and my recovery”—Person-centred dietetic treatment. Across all identified themes was the cross-cutting theme of a shared treatment journey between the dietitian and the individual receiving treatment. These findings support dietitians having a role that is not limited only to the provision of nutrition treatment in ED care and illustrates the importance of dietitians engaging with clients by centring on the individual’s needs and preferences. Further understanding helpful dietetic treatment components and identifying gaps in training is needed to develop these broader roles for dietetic care.

## 1. Introduction

Eating disorders (EDs) are complex mental health conditions with debilitating physical, psychological, and social impacts [1]. The global lifetime prevalence of EDs has been estimated to range between 0.74 to 2.2% in males and 2.58–8.4% in females, and is rising [2]. Recovery from an ED is variable, with approximately 20% of individuals who experience anorexia nervosa (AN) and 10% of people with bulimia nervosa (BN) developing long-standing illness [3,4,5]. Therefore, optimising care is critical to improve the chance of a full recovery.

Dietitians are considered essential members of the multidisciplinary ED treatment team and have an established role in refeeding, as well as providing medical nutrition therapy to address malnutrition and the normalisation of eating patterns [6,7]. However, there is a paucity of empirical evidence to guide the precise content of this medical nutrition therapy or the evaluation of its effectiveness, and current treatment guidelines are based on consensus opinions of expert dietitians and other ED clinicians [8,9]. Alongside this is the need for further research exploring the perspectives of individuals with lived experience of an ED, and what they have valued or found unhelpful as part of dietetic treatments. There has been an increasing call in the ED field for research that integrates lived experience perspectives [10,11]. The benefits of consulting with people with a lived experience throughout the research process include the identification of outcomes and barriers that may otherwise be overlooked, an improvement in content comprehensibility and administration, as well as the provision of greater insight into the interpretation of study results [11]. 

A Delphi study by McMaster et al. reported discrepancies between what ED clinicians and people with lived experience of an ED perceive as essential components of dietetic care [12]. These results highlight that whilst dietitians oversee the provision of nutritional treatment, they can only provide one perspective. Incorporating input from people who experience the treatment is important, as overlooking lived experience in ED research not only risks invalidating these individuals but also risks missing key considerations that may help or harm people seeking or receiving treatment [13]. Further, a lack of literature evaluating evidence-based dietetic interventions and the poor quality of currently available research in this area may partly be due to the lack of consultation with people with lived experience as part of the study design [14].

A previous qualitative systematic review explored the evidence regarding the views and experiences of clinicians, patients, and carers on the role of a dietitian in ED treatment [15]. The results emphasised the desire of dietitians working with EDs for further education in this area to increase confidence, as well as a need for more specific training standards. Whilst this review is current, the findings were focused primarily on the perspectives of dietitians and other mental healthcare providers and did not centre the experiences of individuals who had personally experienced ED treatment. The question remains of what revelations exist in the broader body of literature on people who have experienced ED treatment with a dietitian regarding the overall landscape of dietetic care. Thus, the current meta-synthesis aims to synthesise available qualitative literature to understand the role and value of a dietitian in ED treatment from the perspective of people with lived experience, using a systematic approach.

## 2. Materials and Methods

### 2.1. Study Design

This meta-synthesis employed the method of thematic synthesis [16] and was informed by meta-ethnographic [17] principles. Findings were reported in accordance with the Enhancing Transparency in Reporting the Synthesis of Qualitative Research (ENTREQ) guidelines [18]. The protocol was registered with the international prospective register of systematic reviews (PROSPERO) (ID: CRD42020181868).

### 2.2. Search Strategy

Keywords from three categories relevant to this meta-synthesis were identified. These were words relating to EDs, dietitians, and treatment (full search terms can be found in Appendix B). Keywords from the categories were combined to establish search terms in the following way: ((words related to ED) AND (words related to dietitians) AND (words related to treatment)), with truncation used for key terms. 

The literature search was conducted on 24 October 2022, and included papers from the first date of database publication with no restrictions to capture all available evidence. MedLine, EMBASE, Scopus, ProQuest Dissertation and Thesis, the Cumulative Index of Nursing and Allied Health Literature (CINAHL), and Cochrane Collaboration Database were searched. Google Scholar was also searched for grey literature and the phrases ‘dietitian eating disorder treatment’ and ‘dietitian eating disorder treatment role’ were entered and articles from the first 100 articles from each search were included. The reference lists of included articles were also searched to identify additional relevant articles. The full search strategy can be found in Appendix A.

### 2.3. Study Selection

Citations and abstracts of all articles were exported to COVIDENCE [19] (an online collaboration software platform that streamlines the production of systematic and other literature reviews) and duplicates were removed. One author (YY) then screened all titles and abstracts and excluded papers that did not address the research aims. Published journal articles, guidelines, and grey literature were considered and duplicates missed by COVIDENCE were manually removed. Three authors (PH, JC, CMM) screened twenty per cent of the titles and abstracts each in duplicate to check for consistency. Full texts were then examined by one researcher (YY) for adherence to the inclusion criteria: (1) qualitative or mixed methods research design containing qualitative findings, (2) focused partially or exclusively on the role of a dietitian in treatment for any ED, and (3) were from the perspective of individuals with lived experience of an ED (see Appendix A for detailed inclusion and exclusion criteria). Three authors (PH, JC, CMM) reviewed 25% of full-text articles each for consistency, and discrepancies were resolved by a third author (YY).

### 2.4. Quality Appraisal

The quality appraisal was carried out in duplicate by three authors (YY 100%, PH 50%, JC 50%). Whilst mixed methods papers were included, for the purposes of this meta-synthesis only the relevant qualitative components of the studies were used. The Critical Appraisal Skills Programme (CASP) checklist for qualitative studies [20] was selected as it is specifically designed for the comprehensive evaluation of qualitative studies in terms of their methodological quality, recruitment, data collection and analysis, bias, ethical considerations, findings, and the value of the research. No papers were excluded on the grounds of poor quality. However, papers of higher quality were considered more when interpreting findings and synthesising data. Discrepancies were resolved through discussion with a third author (either JC or PH).

### 2.5. Data Extraction, Translation, and Synthesis

The data extraction process was guided by methods described by Thomas and Harden [16], Shaw (2011) [21], and Sattar et al. (2021) [22]. Three authors independently extracted study characteristics from the papers using a template created on COVIDENCE (YY 100%, PH 50%, JC 50%). One author (YY) read and re-read the included papers to familiarise themselves with the texts. The author then used a standardised data extraction table to extract first-order constructs (i.e., verbatim quotes from the data) and second-order constructs (i.e., author interpretations), and a second author (CMM) checked extracted data to ensure extraction was complete. Using meta-ethnographic principles, two authors (YY and CMM) independently, then collaboratively, created a list of themes from each paper which were then ‘translated’ into one another by comparing similarities and differences and organising them into conceptual categories that “distilled the essence” of ideas presented across papers [23].

Due to a lack of detailed exemplar quotes in all papers, authors analysed the data through an iterative process in line with thematic synthesis whereby conceptual categories were inductively synthesised to create ‘third-order constructs’ or ‘analytical themes’ that aimed to ‘go beyond’ [16] the content of original studies. This did not change the framework or conceptual categories that were developed during translation but allowed for a rich analysis of the data that was embedded within each of the developed themes. This was a recursive process that was first independently conducted by two authors (YY and CMM) and then as a group (YY, CMM, JC, PH, and MKP) until the authors agreed that the new themes were sufficiently abstract to capture the initial conceptual categories.

### 2.6. Reflexivity

All authors are clinicians who have worked with people who have experienced an ED and have also previously conducted qualitative and/or quantitative research pertaining to this area. A more detailed reflexivity statement is provided in Appendix C. The potential for bias given their previous history and expertise was considered throughout the study duration. Reflective logs, regular research supervision, and open discussions were employed to ensure analytical integrity was maintained.

## 3. Results

### 3.1. Study Selection and Characteristics

A total of 15 studies were included in this meta-synthesis. Figure 1 details the PRISMA flow diagram for the search and study selection process.

The study characteristics are detailed in Table 1. The included papers were published between 1995 and 2022 and were conducted across seven countries. All studies included a qualitative methodology component, with data being generated by interviews (*n* = 11), cross-sectional surveys (*n* = 4), focus groups (*n* = 2), field notes (*n* = 2), social media posts (*n* = 1), and session recordings (*n* = 1). A total of 630 participants were included in this review (range 3–310). Participant age was reported in all but three studies [24,25,26] and ranged from 9 to 74. Four studies [26,27,28,29] included adolescent populations (i.e., under 18 years of age), three studies [26,27,29] explored parent or carer perspectives, and 11 studies [24,25,30,31,32,33,34,35,36,37,38] considered only adult-client lived experience perspectives. Four studies [27,29,31,36] used mixed-sex samples, eight studies [24,28,30,33,34,35,36,38] used female-only samples, one study [32] considered male-only samples, and two studies [25,26] did not report sex. Only eight participants were assigned male at birth. The ED diagnoses of study participants included AN (*n* = 8), binge eating disorder (BED) (*n* = 3), BN (*n* = 6), atypical anorexia nervosa (AAN) (*n* = 1), other specified feeding or eating disorder/eating disorder not otherwise specified (OSFED/EDNOS) (*n* = 2), primarily restrictive ED (*n* = 1), ED changed over time (*n* = 1), and three studies [24,27,36] did not specify the diagnosis.

### 3.2. Quality Appraisal

Quality appraisal findings can be found in Appendix A. Whilst all but one study [25] adequately addressed most of the ten areas of the CASP checklist, there was significant variability in the depth of analysis, with results ranging from a superficial or more descriptive analysis to an in-depth and integrative analysis. Ten studies did not adequately consider the relationship between the researchers and the participants. Further, in three studies it was not clear if ethical issues were taken into consideration, and another three studies did not outline a clear and rigorous data analysis process.

### 3.3. Translation and Synthesis

Four themes were generated from the data: (1) “guidance and structure”—Provision of nutrition knowledge and skills; (2) “having all my bases covered”—Dietitians as part of a multidisciplinary team; (3) Challenges in nutritional treatment; and (4) “it was my treatment and my recovery”—Person-centred dietetic treatment. The data also revealed the cross-cutting theme of a shared treatment journey. Figure 2 provides a visual representation of the relationship between the four themes and the cross-cutting theme. Summaries of the included studies and their themes can be found in Appendix A. Further exemplar extracts have also been included in Appendix A.

#### 3.3.1. Theme 1: “Guidance and Structure”—Provision of Nutrition Knowledge and Skills

Seven of the papers identified that aspects of ‘traditional’ dietetic practice relating to nutrition and food were valued by individuals with lived ED experience [26,27,28,29,30,33,35]. 

Subtheme 1(a): Nutrition Education

Education about the science of nutrition and answering “*a lot of questions about chemicals and things like that, that I didn’t know much about*” (p.80) [33] helped individuals to understand the relationship between nutrition and physiology. This was reported to inspire motivation to follow nutritional suggestions made by the dietitian. For example:
*“He [dietitian] was more balanced and helped me learn the difference between I guess (sic) healthy attitudes to wanting to eat well and exercise, versus the eating disorder thoughts wanting to be excessive with exercise and food restriction.”*(P19) [29]

This extract highlights how nutrition education from a dietitian played a role in increasing understanding of how the individual’s eating behaviours and nutritional intake impacted their health, as well as how these could be separated from their ED. This generated reasons for change that included to “*feel better*” (p.79) [33], and to take steps to shift their relationship with food to prevent further “*hurting*” (Mary) [30] the body and “*losing muscle*” (p.79) [33]. In these instances, healthy eating was linked to well-being and care for the body.

Subtheme 1(b): Meal Planning and Monitoring

Logging eating and food diaries were cited by some individuals as important ways to not only facilitate meal planning but also a reflective tool to facilitate change for the individual themselves as illustrated in the following extract:
*“I often log my diet and exercise when I wonder about the reason why I am feeling the way I do. Logging has also enabled me to identify patterns in my eating, the need to adjust my eating, and it prompted me to increase the total amount of food I consume.”*(Chloe) [30]

On the other hand, focus on planning meals for some individuals was noted to increase their focus on their eating in unhelpful ways, for example:
*“I count it... but 30 g of fat. If I added it to everything else in my diet, it’s like, an ordeal, it never was when I was growing up.”*(p.81) [33]

This participant’s experience demonstrates the delicate balance in the role of the dietitian in meal planning, between working with the person towards more flexible eating patterns rather than inadvertently increasing their preoccupation with food restriction. Some individuals identified handing over some responsibility for their eating to the dietitian to be helpful. This assisted in relieving these individuals from feeling the need to consistently think about food, including through the lens of the ED. For example, one participant noted that they “*had lost the feeling of hunger and the feeling of being full and I needed guidance and structure around mealtimes*” (Mary) [30]. In engaging with the dietitian, their focus on food became “*more absent*” (Mary) [30]. This extended to parents and carers who found it helpful to reallocate responsibility for their child’s eating to the dietitian.
*“There was nothing that I could do to help him, so I needed somebody else to be in charge”*(p.488) [27]
*“I guess it was just having somebody else, who wasn’t me, to reiterate the message of what was important. I saw the dietitian as a bit of an ally for me, to help me help her”*(P18) [29]

These extracts exemplify the sense of helplessness felt by carers whose efforts to encourage their loved ones to change their eating patterns recruited them into the sense that “*there was nothing I could do to help him*”. In positioning the dietitian as “*somebody else to be in charge”* of their child’s eating patterns, or “*somebody else… to reiterate the message of what was important*”, these carers were supported by their dietitian to continue care or felt safe to step back from this role and its associated anxieties.

#### 3.3.2. Theme 2: “Having All My Bases Covered”—Dietitians as Part of a Multidisciplinary Team

The importance of having a dietitian as part of a multidisciplinary team (MDT) in ED treatment surfaced substantially in six papers [25,26,29,33,36,38].

Subtheme 2(a): Enhancing Care through a Combined Front

Some participants identified that they held “*very strong belief[s]*” about food and eating due to the ED. Here, having an MDT helped to challenge these thoughts by supporting the individual in the face of the ED through consistent and re-iterative messaging that was experienced by one participant as “*… having all of my bases covered*” (p.7) [38].
*“I think a lot of times when I’m told something […] I immediately get defensive. I don’t fully believe what they tell me. So to have it kind of re-emphasized from another, it’s like having that third-party validation. So like having like them communicate, and the validation is more reassuring [especially] when you have a hard time trusting.”*(Claire) [38]

Having a diverse team of specialised clinicians was identified as important as it meant that individuals could trust the consensus advice from their providers to cover all their “*bases*” and provide comprehensive care. Woodruff et al. (2020) [38] identified that, whilst the roles of different healthcare providers were found to overlap, this overlap was seen as important in reinforcing consistent messaging, although no participant quotes were provided. This was identified as being especially necessary in ED treatment as individuals acknowledged that cognitive impairment as a result of the ED led to struggles in disclosing and retaining important information about their condition or care, for example:
*“... especially when I was really down and out, obviously my brain wasn’t functioning 100%, and so I would forget sometimes to tell someone this. The coordinated care probably helped them [providers] as well … They can tell each other when I wasn’t able to tell them. I think that’s a really important part. And even now, I forget sometimes, “oh yeah I probably should have told you that, but I thought I already said it to someone else,” so the fact that I don’t have to repeat myself is a huge benefit.”*(Isabel) [38]

The importance of communication between team members is illustrated here with a sense of safety being implicit in trusting providers to “*tell each other*” critical information about ED care when participants were not “*able to*”. Another individual even went as far as to state that communication between team members was “*absolutely one of the most important things you can do in eating disorder recovery*” (Kala) [38]. However, communication was reported by some participants to not occur effectively between MDT members.
*“... It was difficult […] I was having to relay my blood results and my weight changes to both the psychologist and dietitian, which I mean it’s frustrating […] but also for an eating disorder patient I don’t think it’s the best idea for them to give me that capacity to lie”*(P11) [29]
*“Even though they both said that they would, I don’t think they ever made contact.”*(Patient 6) [36]

Unspoken in these extracts is the burden faced by these individuals in telling and retelling their stories, challenges, and behaviours to different members of the MDT. This risks situations where problem stories are reinforced or information is missed, potentially leading, therefore, to suboptimal care.

Subtheme 2(b): Challenges with Multidisciplinary Treatment

On the other hand, whilst holistic treatment was mentioned by several participants as important, one individual highlighted how consistent reactionary care from healthcare providers eroded a sense of their own competence.
*“my team that I have right now has given me the space to do that where it’s not this immediate reaction to things, everybody’s letting me test the waters and like figure out for myself, and I think that’s been helpful for me to just … develop some sort of competence that other people aren’t the ones to keep me afloat, that I’m actually doing it, which I think is hard early on in eating disorder treatment, because you do have so many people telling you how to do things, that one of the things that happened to me is that I sort of lost a sense of my own competence.”*(Danielle) [38]

This extract demonstrates how ED treatment, similar to an ED itself, can contribute to the individual losing “*a sense*” of themselves in their own competence, but also how a team approach that centred the individual as the moderator and driver of change was experienced as supporting autonomy. Furthermore, whilst many participants appreciated having multiple practitioners involved in treatment, many noted that treatment from an MDT was challenging to engage with due to time constraints.
*“I know that it is hard to get three or four hours each week to something that you don’t like. Not that I didn’t like, I mean, I love coming, but it’s just something that’s your enemy—to deal with it, you know. Instead of putting it behind and not dealing with it.”*(p.147) [33]

Although this individual identified that finding time was a challenge, implied in their efforts was a motivation to change despite the egosyntonicity and physical effects of the ED. Other participants were unable to continue to see a dietitian as part of their MDT due to financial concerns, for example:
*“There was a point where […] the dietitian got cut because my [EDP], which is the eating disorder plan from the government was like, you’ve used your 20 sessions, so I had to pay full price, and I couldn’t afford that...”*(P08) [29]

ED treatment was identified as imposing a significant financial burden on individuals which can act as a significant barrier to accessing treatment [39]. This is especially concerning as Roots (2009) [26] identified that dietitians were “*repeatedly singled out*” as important healthcare professionals to be involved in ED treatment and that parents were concerned “*when this service was not available*” for their children, although there were no participant quotes.

A lack of treatment availability was also presented as a concern and barrier to treatment. Multiple participants expressed the desire to have more access to a dietitian and wished that they “*could have seen her more often*” (p.83) [33]. This process was made easier by having the team be part of an integrated or multi-modal treatment (i.e., treatments at one centre that involve multiple clinicians working together), which participants identified as helping to facilitate scheduling and communication between clinicians:
*“The fact that … medical/nutritional/practical aspects were very well integrated into my psychotherapy was helpful.”*(p.223) [25]
*“Maybe the most helpful thing was the fact that it was just all together as one program.”*(p.147) [33]

#### 3.3.3. Theme 3: Challenges in Nutritional Treatment

Seven papers found that participants experienced disconnection in the therapeutic relationship with dietitians and reported on ways that some of the participants found the treatment experience with a dietitian to be unhelpful [24,28,29,31,32,34,37].

Subtheme 3(a): Challenges with Working with Ambivalence

Ambivalence towards dietetic treatment was commonly reported and one participant acknowledged that they were *“very aware of the damage I was doing, I could completely understand what I should be doing was very different to what I was doing”* (Alan) [32], but were still unwilling to act. Another *“couldn’t see it [food] as something that could help me”* (Mary) [34]. Others shared experiences with dietitians where their objectives for treatment were not aligned.
*“I really want to help you,” she says. Gosh, how to even begin to explain. I don’t want to eat, don’t want to get fat, food equals fat, I’m afraid you will manage to help me, do you understand, that’s the problem [...] Our goals are different: I want to lose weight and she wants me to get fat. I want a sense of triumph over hunger and my weight, and as far as I’m concerned, she wants me to become weak, she wants me to become someone who will have to eat.”*(p.227) [24]

In these instances, the egosyntonic nature of an ED was apparent, as individuals struggled to see the real effects of the ED and discern problematic aspects of the ED experience. For these individuals, the role of a dietitian in nutritional rehabilitation was seen as a direct contradiction to their desires with some describing the sessions in adversarial terms: for example, “*Every time I meet with my dietitian it is a new battle as to whether the yogurt is going to be 0% or 1.5% fat*” (p.229) [24]. Furthermore, other participants experienced disappointment in themselves or perceived they would be disappointing their dietitian as they struggled to shift their relationship with food, for example:
*“[…] she will be disappointed in me and I will be disappointed in myself”*(p.229) [24]
*“I usually meet with the dietitian twice a week but this week I cancelled. I’m afraid that if I go she’ll ask me to eat in her office (like she did in the past when I couldn’t manage on my own). I want help but my cancellations and distancing myself aren’t letting anyone help me.”*(p.229) [24]

The challenges in dietitians working with an individual’s fear of change were described by some participants as leading to difficulties in accurately disclosing their eating patterns for fear of the perceived consequences and/or the subsequent cancellation of appointments [24,28,34].

Subtheme 3(b): Varied Dietitian Knowledge and Skills in Working with EDs

Some participant narratives highlighted the potential for harm caused by nutritional approaches from dietitians who focused on weight loss for people ‘living in larger bodies’ (preferred terminology for people having a higher weight [40]). Implicit in a number of participant experiences was a lack of dietitian knowledge and skills in the nutritional treatment of EDs for this population. For example, one participant, who identified as living in a larger body, recalled their dietitian stating “*like, okay, whatever*” (Participant 8) [31] when they disclosed that they planned to start using laxatives. Another described the dietitian prescribing restrictive meal plans in recovery to prevent weight gain because they lived in a larger body, despite the participant being weight suppressed:
*“...I [started] actually trying to hit caloric minimums during the day. And, lo and behold, I started gaining weight... and my nutritionist was like, ‘Well, we need to start discussing your caloric intake, because you are gaining weight... we don’t want you to gain too much, and we’re concerned.’ And I felt so betrayed and so upset.”*(Participant 19) [31]

These participants’ stories demonstrate the significant effect that (inadvertent) bias towards weight loss can have on the quality of care provided by dietitians. Implicit in weight bias is the fear of fatness, or fatphobia, leading dietitians to become complicit with the ED, and positing ‘fat’ as bad.
*“I think there are many dietitians out there who will happily fuel eating disorder behavior. I was just really, really lucky that I happened to end up with someone who was a HAES [Health At Every Size ^®^] professional and intuitive eating approach because had she not been, I feel like my restrictive behaviors would have exacerbated tenfold...”*(P20) [29]
*“...When I was my sickest...[my dietitian] was urging me to go to treatment...which was really validating... everyone else in my life...was telling me how amazing [I looked].”*(Participant 8) [31]
*“...Treatment providers [who] incorporated IE* [Intuitive Eating] *and HAES^®^ [are] what led me to...full recovery.”*(Participant 15) [31]

These participant accounts highlight how validation of their struggles with an ED, regardless of body size, and supporting their pathway into treatment were key components of recovery. Participants in larger bodies found that dietitians who validated the ED and took them seriously, regardless of their weight and other ED symptoms, played an integral role in their recovery.

#### 3.3.4. Theme 4: “It Was My Treatment and My Recovery”—Person-Centred Dietetic Care

Nine papers emphasised the importance of dietetic treatment that centred on the individual experiencing treatment [24,25,28,29,30,31,32,33,35].

Subtheme 4(a): Building Trust and Connection

The development of trust with the dietitian was a complex, non-linear process that was further complicated by the nature of the dietitian’s role in ED treatment. Many individuals found the simple fact that a dietitian is “*a dietitian and the one who would raise my intake levels*” (p.68) [28] immediately made them less willing to participate in treatment. In this way, the fundamental role of a dietitian in ED treatment is an inherent barrier to engaging a person in treatment, particularly if they are ambivalent about making changes to their eating behaviour. However, some participants talked about how their dietitian was able to develop and foster a sense of safety when they took the time to connect with them as a person rather than having a single-minded focus on nutritional or medical outcomes [24,33].
*“She did not talk about weighing or about the meal plan. She only said that she wanted to know me better so that we could connect with each other. Until that moment I was sure that dietitians only looked at weight, BMI, and meal plans. I left the clinic with a better feeling, and I felt less worried.”*(p.231) [24]

Dietitians who connected with participants in ways that made them feel like they knew them “*personally*” (p.82) [33] contributed to positive treatment experiences, for example another participant stated that this “*helps me a lot*” (p.82) [33]. This personal connection sometimes took the form of simple gestures such as open and welcoming body language and attitudes (e.g., “*she had a smile, she was much friendlier, she related to me… she was just such a happy positive person*” (p.52) [28]) and at other times involved the dietitian taking a step back to acknowledge the difficult experiences individuals were going through [24,25,28,29].
*“She [dietitian] never made me feel like I was weird, or she never made me feel bad for anything that I was doing or thinking. She was compassionate, had an understanding of, and helped me understand why I was doing the things that I was doing.”*(P17) [29]
*“I entered the session with great fear. The dietitian looked at me and tried to understand why I was shivering. She gave me a glass of water. She closed the opened window. She offered me a hot beverage [...] Slowly, I was able to talk again.”*(p.231) [24]

These extracts highlight the importance of compassionate treatment that focuses on understanding the person’s needs in the moment, not just treatment outcomes, to cultivate a trusting therapeutic relationship. Other participants identified that appropriate self-disclosure from the dietitian, whether in regard to their personal lives or revealing an ED history, allowed individuals to feel that they were *“more than a dietitian just analyzing me”* (p.67) [28,33].
*“I think that she definitely understands where you’re coming from and gives you her personal point of view, like, she gives you examples of what happened to her too. I think that really helps because you really are like, wow, at least somebody else feels like this.”*(p.81) [33]

Here, self-disclosure acted as a means of fostering “*understand[ing]*” and a reduced sense of isolation in the ED experience. Conversely, four papers [24,28,32,37] described experiences where participants lacked an authentic connection with their dietitian, thus leading them to feel like their dietitian “*didn’t like me or even care about*” them (p.54) [28], or they found their dietitian “*quite patronising*” (Alan) [32] or “*very threatening*” (Amy) [37].

Subtheme 4(b): Collaborative Care and Goal Setting

Client-led and collaborative interventions were also perceived as valuable in dietetic treatment across the reviewed studies. Positive relationships were cultivated through the prioritisation of the individual’s sense of autonomy and empowerment of the person, including inviting them to take more responsibility in their treatment. This was especially the case in terms of treatment objectives, such as in this participant’s recount:
*“I think the big thing for my experience was that it wasn’t until a dietitian actually asked me, ‘What’s important to you?’ or ‘What do you want to work towards?’ And my goal was, at that point, to stay out of hospital and she acknowledged that and was like, ‘Okay, well let’s work with that’.”*(P08) [29]

When individuals were able to voice their own goals, this reminded them that this was “*my treatment and… recovery*”. This was particularly experienced when the dietitian positioned themselves as a guide rather than the expert. Another participant highlighted how they respected their dietitian’s willingness to learn from the person’s lived experience (i.e., “*I appreciate that someone who’s worked in the field for quite a while... is still really open to learning and I guess learning not just from formal education and from the professionals, but also from lived experience*” (P22) [29]). This extract highlights the power of the two-way nature of learning in the therapeutic relationship [41]. Furthermore, a sense of self belief was further nurtured when dietitians provided individuals with increased agency in their treatment:
*“I had lots of input into what [my meal plan] was, but of course there were rules around it, [my dietitian] would say you need to have this much or pick one from this category and one from that category, to fulfill your calorie intake”*(p.53) [28]
*“she shows you choices of things...and I’d get to pick the things that I thought I could handle”*(p.67) [28]

Participants recognised that whilst a dietitian is there to provide guidance by setting ‘rules’, it was also important that there was room for the individual to take ownership of their treatment and have a “*choice*” and be able to “*pick*” what they want from treatment. This individualisation of care extended into the participants valuing dietitians who accounted for factors outside of the ED in their care, such as in Reyes-Rodriguez et al. (2016) [35] where the dietitian considered family and interpersonal conflicts as well as cultural values that shaped the individual’s unique challenges with eating and their treatment engagement, although no quotes were provided.

Subtheme 4(c): Dietetic Treatment beyond Nutritional Focus

The role of a dietitian beyond traditional nutrition education was a factor that contributed to positive treatment experiences. Individuals identified that experienced ED dietitians were ones who recognised that EDs were more than simply physiological disorders, but also recognised other contributing factors:
*“I would see someone that has good experience, I wouldn’t see someone that doesn’t specialize because they wouldn’t understand the psychological component...”*(P09) [29]
*“I was afraid it was all going to be just food. Sit down and talk about diet. […] after meeting her I can tell you that it was much more of a personal thing. I was a person and there were more issues involved than just what my body needs. Your body is a machine and you need to give it fuel. I was afraid that someone wasn’t going to consider the fact that there were other issues involved.”*(p.81) [33]

This cultivated a sense of personalised care and being treated as a person rather than the disorder, “*I was a person*” (Subtheme 3a). Participants also highlighted the therapeutic role their dietitians had in their treatment, such as in the following extracts:
*“[my dietitian] took some time to just sit and talk with me about what was upsetting me and so in a lot of ways she functioned a little bit like a therapist and a dietitian...for eating disorders that’s so important because the two [food and emotions] are interlinked, the stuff that is driving you to eat unhealthily are emotional things and the behaviors around the food trigger such an emotional response.”*(p.53) [28]
*“she’s not trained as a therapist but she was willing to do some of that because it is what I needed at that moment”*(p.67) [28]

Here, dietitians who took on the role of both “*a therapist and a dietitian*” were able to address ED concerns more holistically as they recognised the link between food and emotions, which they addressed simultaneously. ED treatment was described as distressing for some participants, and dietitians being able to hold space for the person’s emotional struggles was talked of favourably. Many participants also recognised the need for the dietitian to create, model, and hold boundaries to allow for healing to occur.
*“I experienced an arena where I could expose my feelings completely, and it was like ripping off a bandage and healing the wound.”*(p.512) [30]
*“It was a difficult meeting. She had me eating. She was tough. On the other hand, she hugged me and encouraged me. I came out deciding that I was going to eat, even though it will be hard. [...] It’s far less than what I’m supposed to eat, but it’s a big and hard step.”*(p.230) [24]

These participants appreciated the boundaries that their dietitians held, even when those boundaries were difficult. By balancing “*tough[ness]*” and sensitivity, participants felt their dietitians created an “*arena*” or “*point of sanity*” (p.231) [24] from which they could work through their fears and anxieties. The expectations set by their dietitians also empowered participants “*to learn to have expectations of my own*” (Monika) [30] and create their own meaning of recovery. Conversely, other participants described experiences with dietitians where this balance was not found:
*“My dietitian is really not used to working with girls like me—at least that’s how it seems from her behavior. There are many attempts at “How do you feel about gaining weight?” and “Try and gain some” or “But you’re going to try, ok?” It’s like ... tough when it shouldn’t be, but also soft when it shouldn’t be. It brings me back to the days [during my inpatient treatment] when the dietitian would yell at me if I lost 100 g or not gain anything. [...] What can I do? I need someone to help me build up my strength.”*(p.230) [24]

Implicit in this extract is the dietitian’s anxiety that fuelled her concerns for the person’s weight recovery. For this participant, a previous adverse life experience was triggered, where a dietitian in hospital “*would yell at me*”. This extract illustrates the person’s wisdom regarding themselves that what they needed in nutritional counselling was “*someone to help me build up my strength*”. 

#### 3.3.5. Summary: The Cross-Cutting Theme of a Shared Treatment Journey

Underlying all four themes was the importance of a shared treatment journey experienced by the dietitian and the person receiving treatment.
*“it was the dietitian and I working as a team”*(p.67) [28]
*“She was really empathetic and validating of my existing experiences and was very clear about I’m the expert of my own body and experience and that she was kind of co-working with me, rather [than] directing my input so it was very collaborative.”*(P20) [29]

Participants viewed the dietitian as someone who not only provided holistic care but also walked alongside them in their ED treatment journey, “*co-working*” together as a “*team*”, valuing their expertise on their own life, and in doing so, authentically engaging them in “*collaborative*” care.

## 4. Discussion

### 4.1. Summary of Evidence

This meta-synthesis reviewed 15 qualitative studies that included a total of 630 participants, with the aim of synthesising the current literature to understand what people with lived experience of an ED and treatment perceive as the role and value of a dietitian in ED treatment. Four themes were constructed: (1) “guidance and structure”—Provision of nutrition knowledge and skills; (2) “having all my bases covered”—Dietitians as part of a multidisciplinary team; (3) Challenges in nutritional treatment; and (4) “it was my treatment and my recovery”—Person-centred dietetic treatment. Each theme highlighted an aspect of treatment with a dietitian that was prevalent throughout the included papers, and that was encapsulated by the cross-cutting theme of a shared treatment journey between the dietitian and the person receiving treatment. Whilst separate, synergising all aspects of treatment identified in the themes is required to ensure that effective dietetic care is provided.

The results of this meta-synthesis support the dietitian’s established role in the provision of nutrition-focused interventions, in line with current dietetic practice guidelines [9]. However, participants identified that the benefit of nutrition-focused interventions arose not only from the act of meal planning, monitoring, or nutrition education alone. For some participants, and their carers, food and eating had become burdensome and handing this over to a health professional with expertise in nutrition was an important part of their care [27,29,30]. Other participants reported on the value of nutrition education as providing a lens to view eating to honour their body, as opposed to the punishing perspective reinforced by an ED [29,33]. Similarly, monitoring of eating was more helpful when experienced as an insight-generating activity for the person rather than as a reporting activity to the dietitian [30]. If perceived as the latter, some participants talked about how they were invited to conceal their eating from the dietitian, input false food records, or cancel appointments [24]. Given the egosyntonic nature of an ED [42,43,44], it is not surprising that participants resisted dietetic treatment given that one of the goals of nutritional counselling is eating and weight recovery.

Historically, dietetic practice has been more associated with working with individuals in the ‘action’ stage of change [45]. However, across several included studies, participants did not confine the role of the dietitian to solely helping them with active behaviour change. Participants frequently commented on the significance of the therapeutic relationship with the dietitian in their recovery and reported that a strong therapeutic alliance, where the dietitian sought to understand their experiences and believe in them as a person, was instrumental in their ability to make changes, as consistent with previous ED research [24,25,28,29,33]. These findings are in line with the broader body of work on motivation to change where successful therapeutic relationships, underpinned by emotional validation of the participant and acknowledgement of the difficulty of change, are fundamental to the perceived effectiveness of psychotherapeutic interventions [43,44,46,47]. Participants also identified that the dietitians who specialised in EDs and were upskilled in therapeutic modalities were able to do more than just assist individuals with making changes to their nutritional intake and facilitated a greater understanding of ED experiences. In these accounts, nutritional counselling encompasses both nutritional education and knowledge and other counselling approaches tailored to the experiencing person and their unique context and readiness to change.

An MDT is recommended by ED treatment guidelines for the adequate addressing of nutritional, psychological, and medical facets of care [9,48]. This was endorsed by participants who consistently acknowledged the helpfulness of an MDT that communicated effectively to provide reiterative messaging [38]. However, the participant’s experiences across the studies also highlighted a grey area where the role of the dietitian may overlap with other healthcare professionals, particularly those providing psychotherapy and medical interventions, as participants preferred dietitians who understood the psychological components of ED care and went beyond nutrition-focused treatment [28]. Boundary disputes in healthcare are not new and research suggests that the division of clinical responsibilities in healthcare is not limited to immovable professional roles, but supports dynamic professional boundaries that situationally shift [49]. Similarly, participants reflected that the question was less about who did what, and more about how these roles were negotiated between health professionals and the spirit of collaboration. This indicates that the role of the dietitian and that of other providers may need to be negotiated throughout ED treatment. Therapies such as manualised cognitive behaviour therapy for EDs [50] and family based treatment [51] have no documented role for a dietitian and nutritional counselling is administered by the primary psychological therapist. It is feasible that the dietitian’s expertise in nutrition and pathophysiology, paired with an understanding of psychotherapeutic modalities, may augment such therapies and enrich the therapeutic experience. 

Many participants also commented on the increased burden placed on them when MDT members did not effectively collaborate, and where the burden of communication was placed on the individual [29,36]. Here, transparent, and effective communication within the MDT may help overcome systems issues that may be present in an individual’s social environment and are mirrored in the professional one. Additionally, this line of thinking offers an optimistic outlook via the General Systems Theory [52], namely that the whole can become more than the sum of its parts. With the inclusion of the expertise of a dietitian and thoughtful incorporation of feedback via lived experience, an enhanced and possibly more successful approach to EDs may be possible.

Participant narratives further highlighted the importance of health care professionals engaging in collaborative boundary setting with the person obtaining treatment. The dominant medical discourse centres health practitioners as being the ‘expert authority’, thus leaving those with lived experience “little or no place… in understanding their own symptoms” [53]. However, the ED field has recently seen an increasing presence of lived experience advocacy, with benefits to research including improved study design, implementation, and dissemination [11]. This study extends this by highlighting the potential benefits when practitioners consult an individual on their expertise on themselves, consider their needs and preferences for their treatment, and engage in collaborative goal setting. This is supported by research looking at experiences of AN treatment which similarly identified the desire of those with an ED to have treatments tailored to them, as well as having personal agency in negotiating their treatment [53,54].

Furthermore, lived experience research can reduce the risk of perpetuating the ED [53]. Notable in this study was the importance of dietitians having skills working with people living in a larger body with an ED, who reported on the harm caused by dietitians who encouraged ED behaviours for weight loss purposes, or who did not take the ED seriously [24,28,29,30,31,33]. Participants living in larger bodies expressed that in order to feel safe and understood in their treatment, their dietitian needed to recognise the harms of weight stigma (defined by recent clinical practice guidelines for the management of EDs for people with higher weight as “the disparaging association of higher weight with negative personal characteristics” [40,55]) and tailor their ED treatment to meet their specific needs (e.g., validation of the severity of their ED irrespective of body size and provision of appropriate ED treatment).

### 4.2. Implications for Clinical Practice and Future Research

The current study highlights the importance of dietetic intervention being person-centred and tailored to the individual’s own treatment goals. Whilst this is endorsed in current dietetic practice guidelines, there is a need for more research that explores how dietitians can best deliver evidence-based dietetic treatment components, such as nutrition education and meal planning, which are individualised to the person’s treatment journey. Such research should incorporate input from people with lived experience to enhance treatment acceptability and translation of research into real world clinical practice. 

The results from this review also highlight two important areas of training for dietitians: (1) working with individuals in various stages of change; and (2) dietetic treatment for people with EDs across the weight spectrum. Study findings suggest that with additional training, as part of an MDT, dietitians may provide emotional support alongside a psychologist, although further research is required to identify the scope of this support. Recent dietetic ED practice and training guidelines outline that the dietitian’s role includes “motivation and stages of change and addressing ambivalence and barriers to behaviour change” and suggest that dietitians should have “an awareness of the evidence-based psychological models used in eating disorder treatment” [9]. However, current tertiary education is inadequate in preparing dietitians to provide ED care, and dietitians are left to individually seek training to gain these counselling skills [9,56,57]. Therefore, it would be beneficial for formal dietetic education to incorporate additional training for dietitians working with people who are ambivalent about behaviour change, including approaches such as motivational interviewing [45]. Specifically, these could be offered as elective courses for those interested in working in the ED field. This study also revealed that key aspects of the therapeutic relationship between dietitians and their clients include building trust and connection, believing in the person, demonstrating compassion, firm but gentle boundary setting, and seeing the person beyond their ED, and these could be incorporated into both tertiary education and professional development activities.

The American International Association of Eating Disorders Professionals Foundation (iaedp) offers a credential which allows dietitians to become Certified Eating Disorder Registered Dietitians (CEDRD). The credential provides further specification on the types of skills specialist dietitians can use (e.g., provide therapeutic patient counselling, understanding of techniques such as CBT/dialectal behaviour therapy/acceptance and commitment therapy), but there is little guidance on how and when these skills can be obtained [58,59]. Within the Australian landscape, in 2022, a similar credentialing system was implemented by ANZAED that aimed to formally recognise ED clinicians who had undergone training to meet “minimum standards for the delivery of safe and effective eating disorders treatment” [60]. The purpose of the credentialing system is to help individuals who are experiencing an ED to find and connect with credentialed clinicians who have the appropriate knowledge and experience to treat EDs [61]. Whilst providing much-needed guidance for training requirements, the credentialling system is still in its infancy, and the voluntary nature of the credential, as well as the lack of evidence defining ‘minimum standards’, indicates that more rigorous competency assessments supported by evidence need to be developed.

Current dietetic training also sits within a weight-normative paradigm that inadvertently risks perpetuating weight stigma [40,55]. The normalisation of weight-normative practice can be potentially harmful, especially to individuals presenting in larger bodies for ED care [31,40,55]. Therefore, dietitians providing ED care should obtain further education on how to manage EDs appropriately in this population, and more studies need to explore the effect that weight-bias has in ED care.

Finally, regular clinical supervision and reflective practices for dietitians working in EDs should be considered, similar to that necessitated by psychologists [62]. Given the therapeutic nature of nutritional counselling outlined by participants in this study and the need for a robust and sustainable workforce in the treatment of EDs, dietetic supervision could help to improve clinician and client safety, as well as encourage competent practice. This recommendation is also supported by current dietetic practice and training standards, although the content and structure of dietetic supervision, especially in the area of EDs, is yet to be defined [9,63]. In addition, as many participants commented that their dietitian disclosed their own history of an ED [28,33], clinical supervision would also provide a means of ensuring that self-disclosure is helpful rather than potentially detrimental to the person obtaining treatment. Supervision could also provide a means by which the grey area between the dietitian and other health care professionals can be navigated to ensure effective care.

### 4.3. Strengths and Limitations

This meta-synthesis expands upon the findings of previous meta-syntheses and provides further insight into the perspectives of people with lived experience of ED treatment with a dietitian. The included papers were primarily of good quality. However, the lack of researcher reflexivity consideration in ten of the included studies significantly contributed to the risk of biased study findings. Many of the included studies also reported on the experiences of dietitians as part of an MDT, making it difficult to extricate the data regarding the dietitian from that of other healthcare members. Other sources of potential bias limiting the generalisability of included studies were the lack of consideration of illness duration or severity and the absence of study sample diversity (e.g., only eight male participants were included out of a total of 630, only one study reported on individuals from gender-diverse backgrounds, and many studies did not disclose the race of included participants). Furthermore, the interventions provided by the dietitians were not described in many of the studies. Hence, participants’ experiences of the dietetic intervention may have varied significantly due to differences in the treatment provided. Future research would benefit from including samples that are representative of studied populations. Further, the inconsistencies in analysis depth (from descriptive to integrative) could have skewed the results of this current review. Qualitative research in the future needs to extend beyond description and provide a more in-depth analysis of participant experiences of dietitian ED treatment.

A key limitation of this review was the exclusion of studies that were not reported in English, which could have led to relevant literature being missed. Similarly, whilst a second reviewer screened a proportion of articles at each stage, there is the possibility that a single reviewer may have overlooked some relevant publications. Furthermore, a significant number of transcript extracts could not be included in this review, and the generated themes were influenced by the researcher’s focus on the dietitian’s role in ED treatment. Despite these limitations, this review exhibited many strengths including registration of the review protocol with PROSPERO, the rigorous search strategy, which included the searching of grey literature, and the comprehensive inclusion criteria. The data extraction and coding processes were also conducted independently by multiple researchers which reduced the risk of bias.

## 5. Conclusions

This meta-synthesis supports the role of a dietitian extending beyond solely helping people with active eating behaviour change in ED treatment. Dietitians are well placed to enhance treatment engagement as well as hold space for a person’s emotional struggles relating to an ED. Integral to this is the dietitian’s ability to provide compassionate and individualised care that embodies the spirit of collaboration, both with the individual experiencing an ED, and the MDT. However, more research is required to understand what additional training dietitians require to safely step into these broader therapeutic roles. Further research that explores the composition of helpful dietetic treatment components is also needed to aid the development of clearer dietetic ED practice standards and guidelines. Such work would benefit from the inclusion of co-design principles that centre lived-experience perspectives.

## Figures and Tables

**Figure 1 behavsci-13-00944-f001:**
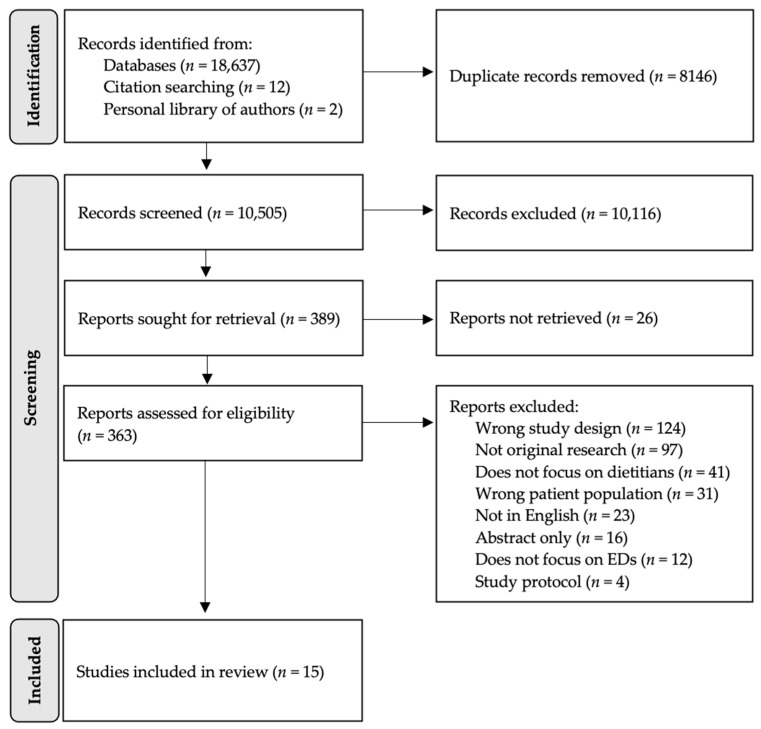
PRISMA flow diagram outlining identification and selection of included articles.

**Figure 2 behavsci-13-00944-f002:**
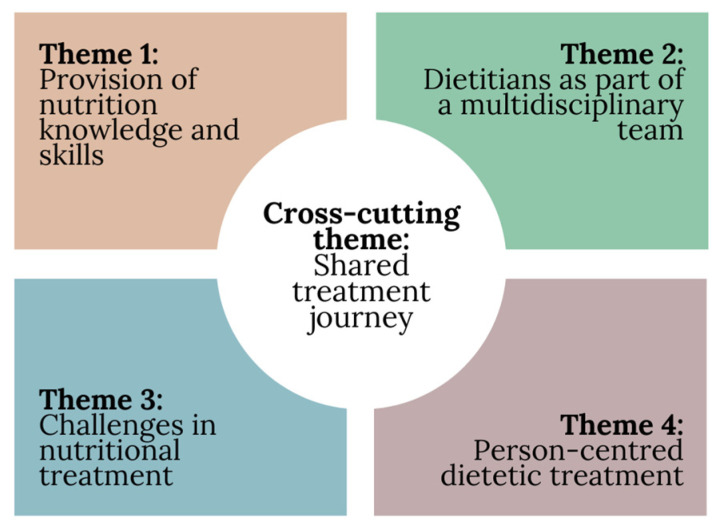
Summary of included themes and cross-cutting theme.

**Table 1 behavsci-13-00944-t001:** Characteristics of included studies.

Author (Year), Country	Study Aim (s)	Recruitment	Participants	ED Diagnosis (n)	Treatment	Data Collection	Method
Bakland (2019), Norway [30]	To explore important aspects of the patients’ own perceived benefits (or not) of the treatment as well as their experiences related to the various treatment components.	Treatment setting	Participants *n* = 15 %Female = 100 Age 19–42 years (mean NA)	BED (9) BN (6)	Setting: outpatient Physical exercise and dietary therapy (PED-t) conducted by physical exercise therapists and dietitians, with CBT.	Semi-structured interviews	Thematic analysis
Blumenthal (2020), USA (Thesis) [31]	To discover how individuals with a history of (or current) AAN have experienced weight bias from nutrition providers during their illness, treatment, or recovery by describing their perceptions of the attitudes, treatment approaches, and phenotypes of the nutrition providers who treated them.	Community sources, online, treatment setting, snowballing	Participants *n* = 20 %Female = 85 %AFAB-NB = 10 %AMAB-F = 5 Age 22–74 years (mean 37.6)	AAN (20)	Setting: inpatient and outpatient Any treatment with a nutrition provider.	Semi-structured interviews, cross-sectional survey	Thematic analysis
Bravender (2017), Switzerland [27]	To describe parent and patient impressions of inpatient medical stabilisation, both qualitatively through directed interviews, and quantitatively through parent and patient evaluation of specific components of an inpatient EDs medical stabilisation protocol.	Treatment setting	Patients *n* = 23 Parents *n* = 32 %Female = 81 %Male = 9 Gender/Sex parents NA Age patients 9–21 years (mean 14.9) Age parents NA	NA-patients were admitted under inpatient protein calorie malnutrition protocol	Setting: inpatient Medical stabilisation through nutritional refeeding. Clinicians involved included dietitians, massage therapists, nursing staff, patient care associates, and physicians.	Semi-structured interviews, cross-sectional survey	Descriptive theme identification
Darden (2017), USA [28]	To investigate the patient’s perception of the facilitators and barriers to forming a therapeutic working alliance with the RDN during ED treatment.	Online	Participants *n* = 7 %Female = 100 Age 15–51 years (mean 26.6)	AN (4) BN (3)	Setting: inpatient and outpatient Had to have completed an ED treatment program and have had treatment from a dietitian.	Semi-structured interviews	Thematic analysis
Elran-Barak (2022), Israel [24]	To explore the perspectives of women with an ED regarding their nutritional counselling.	All social media posts meeting the inclusion criteria were included.	Social network users: Study 1 *n* = 82; Study 2 *n* = 14 %Female = 100 Age NA	NA	Setting: NA Treatment involving nutritional treatment.	Posts made on a medical social network website	Phenomenological analysis Thematic analysis Content analysis
Heafala (2022), Australia [29]	To explore perspectives on receiving dietetic care for EDs in a primary care setting.	E-newsletters, ED organisation websites	Participant with lived experience of ED n= 21 Carers of people with ED *n* = 3 %Female = 96 %Male = 4 Age 15–54 years (mean NA)	AN (12) AN and BN (3) OSFED (2) Changed over time (2) Primarily restrictive (2)	Setting: outpatient Participants had to have seen a dietitian in a primary setting.	Semi-structured interviews	Thematic analysis
Lyons (2018), UK [32]	To explore the lived experiences of men who have, or have had, an ED in the form of anorexia or an atypical variant (EDNOS).	Online, ED charity volunteer database	Participants *n* = 7 %Male = 100 Age 23–34 years (mean 28.29)	AN (7)	Setting: inpatient and outpatient Varied dietitians/nutritionist, counselling, hypnotherapy, help lines and accident and Emergency NB. All men interviewed had sought treatment, all but one had received medical.	Narrative interview (with structured questions)	Narrative analysis
Marek (1995), USA [33]	To evaluate an integrated treatment program for college women with eating problems.	Flyers, presentations, announcements, and newspaper advertisements	Participants *n* = 11 %Female = 100 Age 18–22 (mean NA)	AN (2) BN (6)	Setting: outpatient Bi-weekly dietitian appointments, bi-weekly family therapist appointments, weekly group therapy sessions by dietitian and therapist.	Semi-structured interviews, focus groups, field notes	Grounded theory
Munro (2014), UK [25]	To describe the service model and present preliminary evidence that begins to answer if intensive community treatment can avoid or minimise the use of inpatient care, if treatment for severe AN can be delivered safely in the community and be acceptable to patients, and if it is cost-effective.	Treatment setting	Patients *n* = 33 Gender/Sex NA Age NA	AN (33)	Setting: outpatient Intensive therapy service involving schema therapy, dietetic treatment, meal support, social support, and medical monitoring.	Cross-sectional survey	NA
Petry (2017), Brazil [34]	To comprehend how women in recovery from AN feel and think about their eating behaviour both during and after this ED experience.	ED service	Patients *n* = 3 %Female = 100 Age 21–24 years (mean 22.7)	AN (3)	Setting: outpatient Monthly individual psychological, nutritional, and psychiatric treatment + weekly group psychotherapy.	Semi-structured interviews	Phenomenological analysis
Reyes-Rodriguez (2016), USA [35]	To examine the content of nutritional sessions that participants of PAS Project- “Promoviendo una Alimentación Saludable” (Promoting Healthy Eating Habits) received as part of a small pilot clinical trial for EDs.	Treatment setting	Participants *n* = 18 Dietitian * *n* = 1 %Female = 100 Age 18–50 years (mean 38.5)	BED (5) BN (6) EDNOS (7)	Setting: outpatient Up to three nutrition sessions with a dietitian. A flexible personalised approach (not manualised) used.	Qualitative analysis of the nutritional therapy session recordings	Grounded theory
Roots (2009), UK [26]	To assess young persons’ and parents’ satisfaction with CAMHS outpatient, specialist outpatient and inpatient treatment received in a large randomised controlled trial.	ED service	Survey: Adolescents *n* = 160; Parents *n* = 150 Focus group: Adolescents and parents *n* = 21 Gender/Sex NA Age NA	AN (160)	Setting: inpatient and outpatient Inpatient admission, specialist outpatient therapy, and treatment as usual in the community.	Cross-sectional survey, focus groups	Thematic analysis Content analysis for part 1
Taylor (2021), USA [36]	To provide further insight into the unique challenges involved in healthcare collaboration when using technology. Second, to offer suggestions about designs that might better support, for all stakeholders, the collaborative nature of ED treatment.	ED recovery websites and social media groups	Patients *n* = 9 Clinicians * *n* = 10 %Female = 89 %Male = 11 Age 18–45 years (mean 28)	NA	Setting: NA Participants had to have been receiving treatment for at least 6 months.	Semi-structured interviews	Grounded theory
Thompson (2007), USA [37]	To explore how women with BED perceive the value of social support in their recovery processes.	Community sources, ED service, clinicians from outpatient ED service	Patients *n* = 10 %Female = 100 Age 31–53 years (mean 40.7)	BED (10)	Setting: outpatient Minimum 1 year in individual therapy/counselling.	Semi-structured interviews, demographic questionnaires, field notes	Phenomenological analysis
Woodruff (2020), USA [38]	To understand the patient experience of women receiving coordinated, multidisciplinary treatment for an ED in an outpatient setting, employing a qualitative methodological approach.	ED service	Patients *n* = 12 %Female = 100 Age 18–42 years (mean NA)	AN (7) BN (3) Unspecified ED (2)	Setting: outpatient Collaborative team consisting of one physician, one nurse practitioner, 6–8 therapists and dietitians. Weekly to less-than-monthly frequency depending on clinical needs and financial resources.	Semi-structured interviews	Grounded theory

Abbreviations: AAN (atypical anorexia nervosa); AFAB-NB (assigned female at birth—non-binary); AMAB-F (assigned male at birth—female); AN (anorexia nervosa); BED (binge eating disorder); BN (bulimia nervosa); CAMHS (child adolescent mental health service); CBT (cognitive behavioural therapy); ED (eating disorder); EDNOS (eating disorders not otherwise specified); RDN (registered dietitian nutritionist); NA (not available); NB (nota bene/note). * Clinician responses were not used in this current paper and were not included in the total participant number.

## Data Availability

Data are contained within the article and Appendix A.

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
