# Peer review of "“I Need Someone to Help Me Build Up My Strength”: A Meta-Synthesis of Lived Experience Perspectives on the Role and Value of a Dietitian in Eating Disorder Treatment"

_behavsci, 2023, doi:10.3390/bs13110944_

Round 1

Reviewer 1 Report

Comments and Suggestions for Authors

Thank you for inviting me to read this manuscript. I found it well written and of high quality.

The abstract

Well structured, easy and clear to read. I have one comment for you to reflect on. In the abstract you use “people” and “person” interchangeable, and later in the article you also use individuals for persons with ED. In the included articles the participants are recruited from different settings, and not all are patients. I would therefor prefer “person”, and to use it consistently, but you might have some idea behind this.

Introduction

This is also very clear and condensed and still consist of all relevant information. The study is well motivated by the included references that also support the aims.

Materials and methods

The methods used in this meta-synthesis is clearly described, the results reported according to guidelines and the study was preregistered, anything you could ask for. The process is clearly described.

Under the heading reflexivity all authors preunderstanding is described, however their profession is not presented. In this case it would have been of interest to know their professions and if they had worked with psychotherapeutic methods or not, and if any of them was a dietarian. This curiosity is based on my experience of the sometimes problematic interest in talking about food and exercise among patients with AN.

Results

The results are well described. However, I wonder about the inconsistency in the participants labelling: “p.79”, “P19” or name as “Mary”. Moreover, there are a bit of discussion within the result section, including references to other articles, for example line 288-293, or 360-365. I prefer to be more conservative and only present the results in this section.

Discussion

Also, this part is well written and all results are discussed. However, I miss a discussion about if the participants experienced the same intervention, since there is no gold standard dietarian intervention. Maybe, the dieticians provided very different things, and the lived experience might have differed significantly between different participants.

Reviewer 2 Report

Comments and Suggestions for Authors

This is a systematic synthesis of articles on the role of dietitians in treatment teams of eating disorders. The authors have prepared the manuscript relatively well. There is one comment for the authors’ consideration. 

In the discussion about clinical practice and future research and conclusion, it would have been helpful to see recommendation of the “ideal” role dietitians may plan during the team-based treatment journey for ED patients. Even though various points were made about the need for training in addressing motivation and stages of change type of behavioral/emotional aspect, a clear recommendation of how such training may be incorporated into formal trainings would be helpful. 

Reviewer 3 Report

Comments and Suggestions for Authors

Thank you for the opportunity to review this meta-synthesis, which looks at patient experiences of the involvement of dieticians in the context of eating disorder. The paper is well written and provides a clear rationale for the review. Methods are robust, reproducible and justified. For these reasons, I think the review is worthy of publication but I have some suggested amendments below which I believe would considerably improve the readability and transparency of the paper.

Abstract: rationale presented and some sense of results, but typically for review paper I would also expect to see some of the results for the review process itself (i.e. how many search results were there etc.).

Introduction:

l.50 – 51 reference to a recent review is a bit confusing – it sounds like the authors are justifying the current study but this is a meta synthesis and therefore qualitative – the reference to the need for further quantitative research therefore feels a bit misplaced.

l.54 I think it is important not just to mention what patients have valued but also to identify elements of dietary approaches with which they might have struggled.

l.62 – 69 would benefit from citations.

Good justification for novelty of review at the end of the Introduction.

Methods:

Can you clarify date restrictions? I’m not sure if the current phrasing is suggesting that there were no restrictions? If so, please state this more clearly. Also justify.

Taking the first 100 from Google Scholar seems a little arbitrary (although I appreciate the point that Google Scholar’s results are somewhat endless and lose relevance, and also that the authors have ensured the searches are replicable by including this information) – can you justify the decision?

Were backward and forward citation searching both applied?

Inclusion and exclusion criteria need to be much more explicit.

Can you provide an indication of inter-rater reliability at the screening stage?

Findings

Why were there 389 records sought for retrieval? It looks like there should be 288. (i.e. 10505 – 10217).

Similarly, there looks to be an error when including reasons for exclusion. There are 363 studies assessed, 333 look to be excluded (i.e. reasons provided) but only 15 studies are included, so what happened to the other 15?

l.205 – 7 – this point feels like it should belong in the Discussion when synthesising the findings rather than the very beginning of the Findings (i.e. it is creeping into clinical recommendations rather than findings from the studies).

l.233 onwards – I think somewhere there should be a caveat that food diaries and logging can also have a negative effect. This comes later so perhaps a rearrangement of the text would address the issue.

Frequent mention of MDT from line 272. I think given that the section is all about this it would be helpful to spell out the abbreviation in full.

Discussion:

l.700 not clear who the ‘larger body’ is – could this sentence be revisited for clarification please?

Further confusing is l.702 reporting ‘participants living in larger bodies’. This is odd phrasing that has not been used up to this point. It also implies that the review has focussed only on those who are trying to gain weight, but given the inclusion of many different EDs, this is not necessarily true. These sentences would benefit from some re-phrasing.

l.721 onwards. Again, I think these sentences make some assumptions that might not be reasonable; specifically, the statement that dietician training is inadequate to provide ED care. While that may be true, dieticians train to implement many treatments and programmes other than ED care, and it is not unreasonable to think that in order to be qualified to work with an ED population they should seek out further training. Suggest re-phrasing to avoid judgment/criticism of the existing training process and to recognise instead that the training in and of itself would not equip a dietician to work with this complex patient group.

L.733 Please explain what a credentialling system is. It would also be useful to outline whether similar models are applied in other countries as this is not an Australian journal and the review did not focus only on studies conducted in Australia.

l.753 I’m sorry if I’ve missed this but I’ve been back to look and cannot find reference to the fact that dieticians were disclosing their own historical EDs in the Findings, only mentioned in the Discussion. The Discussion should not introduce new material in this way so please address or confirm where in the Findings this is mentioned.
